# Elucidating SNP-Based Population Structure and Genetic Diversity of *Bruguiera gymnorhiza* (L.) Savigny in Thailand

**Panthita Ruang-areerate** [1], **Chutima Sonthirod** [1], **Duangjai Sangsrakru** [1], **Pitchaporn Waiyamitra** [1], **Chatree Maknual** [2], **Poonsri Wanthongchai** [2], **Pranom Chomriang** [2], **Wirulda Pootakham** [1] and **Sithichoke Tangphatsornruang** [1,*]

[1] National Omics Center, National Science and Technology Development Agency (NSTDA), Pathum Thani 12120, Thailand

[2] Department of Marine and Coastal Resources, The Government Complex, Chaengwatthana Rd., Thung Song Hong, Bangkok 10210, Thailand

\* Correspondence: sithichoke.tan@nstda.or.th

**Abstract:** *Bruguiera gymnorhiza* (L.) Savigny is one of the most important and widespread mangrove species in the Indo-West Pacific region. Here, the population structure and genetic diversity of *B. gymnorhiza* along the coastlines of Thailand were examined. A total of 73 *B. gymnorhiza* accessions in 15 provinces were sequenced using RAD-seq to generate their SNPs. Based on the high-quality SNPs, the topology of the maximum likelihood phylogenetic tree clearly presented two genetically distinct groups corresponding to two geographic regions, the Gulf of Thailand and the Andaman Sea coasts. The results for the population structure provided by STRUCTURE and PCA also showed two main genetic clusters and their genetic admixture. A moderate genetic diversity was observed among the accessions, with average observed and expected heterozygosity values of 0.397 and 0.317, respectively. A high genetic differentiation ($F_{ST}$ = 0.16, $p < 0.001$) between the two subpopulations was significantly found. An analysis of molecular variance revealed 83.95% of the genetic variation within populations and 16.05% of the genetic variation among populations. A high genetic variation within the populations and admixture may facilitate adaptation to local environments and climate changes. These results provide important information on the population genetic structure and genetic diversity of *B. gymnorhiza* in Thailand for further mangrove management.

**Keywords:** *Bruguiera gymnorhiza*; Rhizophoraceae; single-nucleotide polymorphism; population structure; genetic diversity; mangrove forest; RAD-seq

## 1. Introduction

Mangroves are halophytes (salt-tolerant plants) that have adapted to the extreme conditions of intertidal zones in tropical and subtropical regions. They play important roles in coastal ecosystems, including coastal protection; carbon sequestration; metal absorption; nursery support; and food, wood, and medicine production [1,2]. Remarkably, mangrove forests absorb three to four times more carbon dioxide than other tropical forests [3]. Global mangrove forests have continually declined over several decades due to aquaculture, conversion to agriculture, overexploitation, and urban development [4,5]. The global area of mangrove forests was estimated at approximately 152,604 and 147,359 km² in 1996 and 2020, respectively [6]. In Thailand, mangrove forests can be found mostly along the coastlines of the Andaman Sea and the Gulf of Thailand. Thailand's mangrove forests lost more than 50% of their areas between 1961 and 1996 due to the expansion of shrimp and salt farms [7,8]. From 1996 to 2020, the areas of Thailand's mangrove forests were estimated at approximately 2598 and 2528 km², respectively [6]. Anthropogenic activities and climate change cause the reduction and fragmentation of mangrove forests, leading to the loss of mangrove genetic diversity. Therefore, the study of the genetic diversity of mangrove

species is now essential, as understanding the population structure and remaining diversity is a key consideration for the conservation and management of this important coastal forest ecosystem.

*Bruguiera gymnorhiza* (L.) Savigny (large-leafed orange mangrove) is a true mangrove species belonging to the family Rhizophoraceae. It is an important mangrove species in the Indo-West Pacific (IWP) region [1,9], and it is widely distributed from Africa's eastern coast through Asia to subtropical Australia and Oceania [1]. *B. gymnorhiza* has the ability to adapt to various conditions, such as a wide range of sunlight shade, saline soil, and water [1,10]. Its barks, leaves, fruits, and roots provide many medicinal properties that have been used in traditional medicines in several countries for treating common diseases, such as diarrhea, fever, malaria, and eye disease [11–13]. This mangrove is viviparous, producing seeds germinating on mother plants [1]. *B. gymnorhiza* is reported to possess a mixed mating system, mainly outcrossing with bird pollination [9,14–16]. *B. gymnorhiza* and other *Bruguiera* species have knee roots that emerge from the ground to exchange gases in oxygen-poor sediments. Among all *Bruguiera* species, *B. gymnorhiza* has the largest leaves (up to 25 cm in length) and solitary large flowers with calyces mostly pinkish to reddish [1,9,16,17]. Based on morphology, *B. gymnorhiza* is difficult to distinguish from other *Bruguiera* species, especially *Bruguiera sexangula* [1,18]. Currently, the whole and chloroplast genomes of the species have been sequenced [19–21]. Chloroplast sequence regions have been used to identify closely related species of mangroves and other plants [20,22,23]. Notably, molecular markers, such as nuclear genomic regions, chloroplast sequence regions, and random amplified polymorphic DNA (RAPD), can clearly distinguish *B. gymnorhiza* from *B. sexangula* [19,20,24,25].

For assessing the population structure and genetic diversity of plant species, the use of molecular markers is one of the most precise and efficient methods [26,27]. Among molecular markers, single-nucleotide polymorphisms (SNPs) have emerged as the most prevalent type of molecular marker in the postgenomic era due to their high density across genomes and rapidly falling costs [26–28]. SNPs, while individually less informative than microsatellites, are typically used in very large panels comprising hundreds or thousands of loci: they are therefore typically more precise than microsatellites in estimating genetic diversity, and they allow for the consideration of local adaptation [29,30]. SNP markers have been successfully used to evaluate the population structure and genetic diversity of mangrove species [31–34]. For *B. gymnorhiza*, the population structure and genetic diversity have been examined using various molecular markers, such as nuclear regions, chloroplast regions, RAPD, inter-simple sequence repeats, and microsatellites [14,25,35–40]; however, they have never been identified based on SNP markers. According to previous studies, the population structure of *B. gymnorhiza* was observed to be divided into two genetic clusters: the east and west clusters of the Malay Peninsula [37]. A low genetic diversity in *B. gymnorhiza* was found in the southwestern islands of Japan (Okinawa and Iriomote islands), South China, India, Indonesia, Malaysia, Micronesia, Thailand, and the IWP regions [14,35–37,39,40]. In contrast, a medium-to-high genetic diversity in *B. gymnorhiza* was observed in Rekawa of Sri Lanka [25] and in the Indian Sundarbans [38].

To date, there has been no study on the genetic diversity and population structure of the natural population of *B. gymnorhiza* along the coastlines of Thailand. In this study, we use high-quality SNP markers to assess the population genetic structure and diversity of 73 *B. gymnorhiza* accessions collected from 15 provinces along Thailand's coasts. These results provide information on the level of genetic variation in *B. gymnorhiza* in Thailand to support mangrove forest conservation and management.

## 2. Materials and Methods

### 2.1. Sample Collection and DNA Extraction

In 2020 and 2021, the young leaves of 73 *B. gymnorhiza* accessions were collected in 15 provinces of Thailand (Table S1). A total of 47 sampling sites in 11 provinces (Chonburi: CBI, Chachoengsao: CCO, Chumphon: CMP, Nakhon Si Thammarat: NST, Phetchaburi:

PBI, Prachuap Khiri Khan: PKN, Rayong: RYG, Samut Songkhram: SKM, Samut Sakhon: SKN, Surat Thani: SNI, and Trat: TRT) are located on the Gulf of Thailand, and 26 sampling sites in 4 provinces (Phang-nga: PNA, Ranong: RNG, Satun: STN, and Trang: TRG) are located on the Andaman Sea. The geographic locations of the sampling sites are presented in Figure 1, developed using QGIS software v3.24.2 (http://www.qgis.org, accessed on 6 December 2022).

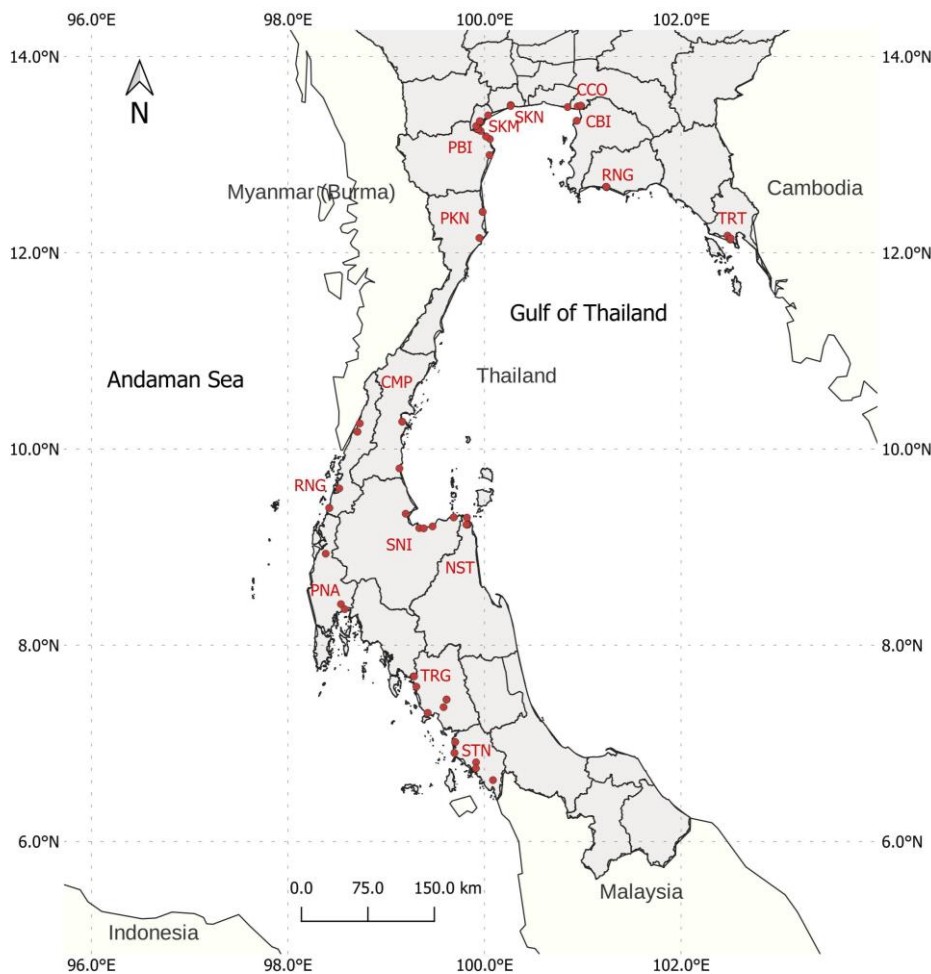

**Figure 1.** Geographical locations of 73 *B. gymnorhiza* (L.) Savigny accessions in Thailand. Red dots and texts represent sampling sites and abbreviations for provinces, respectively.

The genomic DNA of the young leaves in each accession was extracted using the standard CTAB method [41] followed by a cleanup using a DNeasy Plant Mini Kit (Qiagen, Hilden, Germany). A Qubit fluorometer (Thermo Fisher Scientific, Waltham, MA, USA) and a Qubit dsDNA BR Assay kit (Invitrogen, Waltham, MA, USA) were used to measure the concentration of the extracted genomic DNA.

### 2.2. RAD-Seq Library Construction and RAD-Seq Sequencing

One of the reduced-representation library sequencing methods is restriction-site-associated DNA sequencing (RAD-seq), which enables the identification of a large number of genome-wide markers across numerous individuals [42,43]. RAD-seq has been shown to be a cost-effective and efficient method for SNP discovery and genotyping, especially in non-model organisms where whole-genome sequencing may be impractical [42,44,45]. RAD-seq libraries were constructed with ~1 µg of DNA using the MGIEasy RAD Library Prep Kit Instruction Manual (MGI Tech). Briefly, the TaqI restriction enzyme was used to cut genomic DNA. DNA fragments were obtained and ligated with unique barcoded

adapters. Pooled RAD-seq libraries were sequenced to generate 150 bp paired-end reads using the MGISEQ-2000RS sequencing platform following the manufacturer's protocol.

### 2.3. SNP Identification and LD Pruning

The sequence reads for each accession based on each unique barcode were aligned with the published whole genome of *B. gymnorhiza* (BioProject accession number PRJNA725949) [19] using BWA [46]. Variances were called using GATK v4.1.4.1 with the HaplotypeCaller mode [47]. SNPs were identified and filtered using BCFtools v1.12 [48] and VCFtools v0.1.16 [49] using the following criteria: (1) base quality scores > 30, (2) coverage depths between 10× and 200×, (3) missing data ≤ 5%, and (4) a minor allele frequency ≥ 0.05. The file format of the SNP data was a variant call format (VCF) file that was converted to other file formats for further analysis using PGDSpider v2.1.1.5 [50]. After filtering, SNPs in strong linkage disequilibrium (LD) were pruned out to reduce the effects of LD on genetic variance using PLINK with a variant pruning tool (–indep-pairwise 50 5 0.5) by defining a window of 50 SNPs, removing 1 of an SNP pair if $r^2 > 0.5$ and then shifting the window by 5 SNPs and repeating the procedure [51].

### 2.4. Phylogenetic Analysis

Multiple sequence alignment and the best substitution model for the SNP data were performed using MUSCLE to find the best DNA/protein models embedded in MEGA X, respectively [52]. A maximum likelihood (ML) phylogenetic tree was preformed based on the best-fit nucleotide substitution model, Tamura-3-parameter (T92+G), and 1000 bootstrap replicates (estimating branch support) using MEGA X [52]. An unrooted tree was visualized using iTOL v6 [53].

### 2.5. Population Genetic Structure Analysis

The population structure of *B. gymnorhiza* was identified using two approaches: a Bayesian clustering approach and a principal component analysis (PCA). For the Bayesian clustering approach, the population structure pattern was inferred using the STRUCTURE program v2.3.4 with 20 runs per *K* (the number of subpopulations), 10,000 burn-in period iterations, and 10,000 MCMC iterations for *K* = 1–16 (based on 15 provinces of sampling sites) under a genetic admixture model [54]. To estimate the appropriate *K* value, the delta *K* value (Δ*K*) based on the formula defined by Evanno et al. (2005) [55] was calculated using the STRUCTURE Harvester program v0.6.94 [56]. The average cluster membership proportions for the 1000 repetitions of a specific *K* value were calculated using CLUMPP v.1.1.2 [57]. For PCA, the eigenvalues of the SNP makers (the proportion of variance explained by SNPs) were generated from the VCF file using PLINK v1.9 [51]. The first two PCs scores were visualized using R software v3.3.4 with the package ggplot and the library tidyverse [58].

### 2.6. Genetic Diversity Analysis

Genetic diversity parameters, including gene diversity (heterozygosity), polymorphic information content (PIC), and minor allele frequency (MAF), were calculated using PowerMarker v.3.25 software [59]. In addition, the levels of genetic variation, including the average number of alleles per locus (*N*a), the effective number of alleles per locus (*N*e), Shannon's information index (*I*), observed heterozygosity (*H*o), expected heterozygosity (*H*e), the proportion of polymorphic loci (PPL), and the fixation index (*F*), were calculated for the clusters (subpopulations) defined by a population structure analysis using GenAlex v.6.502 [60]. Furthermore, an analysis of molecular variance (AMOVA) and population differentiation ($F_{ST}$) was carried out on the clusters using ARLEQUIN v.3.5 [61].

## 3. Results

### 3.1. SNP Identification and Characterization

A total of 1,051,297,454 raw reads for 73 *B. gymnorhiza* accessions were obtained, with an average of 14,401,335 reads per accession (Table S2). An average of 13,032,114 reads (90.49%) were mapped onto the reference genome of *B. gymnorhiza* (Table S2). A total of 2,831,356 SNP loci based on the 73 accessions were initially identified. According to the SNP criteria mentioned in the Materials and Methods Section, the SNPs were filtered to obtain 2887 high-quality SNPs. After removing the SNPs in strong LD with a threshold of $r^2 > 0.5$ (default), a total of 1519 SNP loci remained (Table S3).

The distributions of the values for SNP diversity, PIC, and MAF of the 1519 SNP markers in the 73 *B. gymnorhiza* accessions are presented in Figure 2 (Table S3). The diversity of these SNP markers ranged from 0.10 to 0.50, with a mean of 0.29 (Figure 2A). The PIC values for the SNP markers varied from 0.09 to 0.38, with a mean of 0.24, indicating low-to-moderate polymorphism information (Figure 2B). The distribution of MAF varied from 0.05 to 0.50, with a mean of 0.21 (Figure 2C). Approximately 31% (467 SNPs) of these SNPs were low MAF (0.05–0.10), revealing an excess of minor alleles at low frequencies (Figure 2C).

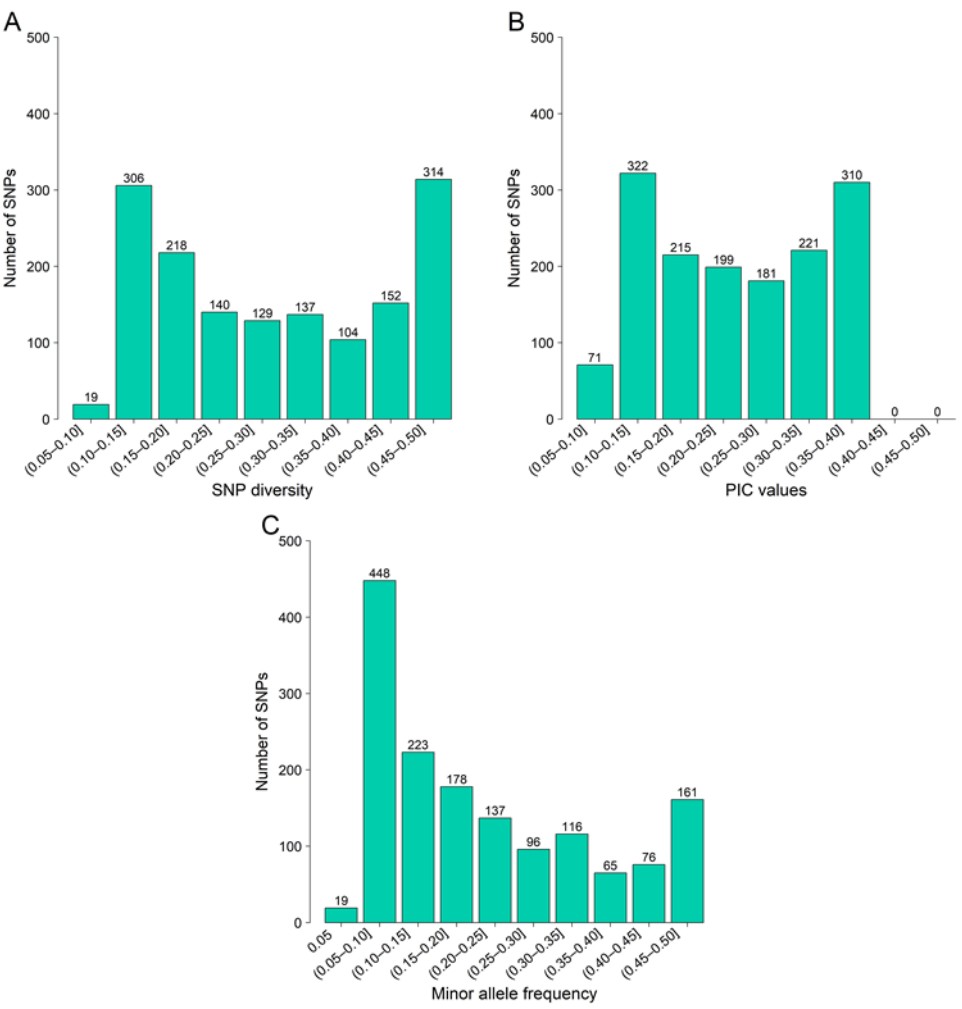

**Figure 2.** Distribution of 1519 SNP markers estimated for all 73 *B. gymnorhiza* accessions. (**A**) SNP diversity; (**B**) polymorphic information content (PIC); (**C**) minor allele frequency.

### 3.2. Phylogenetic Tree

The unrooted ML tree depicting the genetic relationships among the 73 *B. gymnorhiza* accessions based on the 1519 SNP markers showed 2 genetically distinct groups (Figure 3).

The longest branch on the ML tree was the stem connecting the accessions from the Gulf of Thailand (blue) and the Andaman Sea (green). It had bootstrap support values greater than 90%. Four accessions (CCO_03, SNI_01, SNI_03, and SNI_05) from the Gulf of Thailand coast were located between the two genetic groups, suggesting admixed accessions.

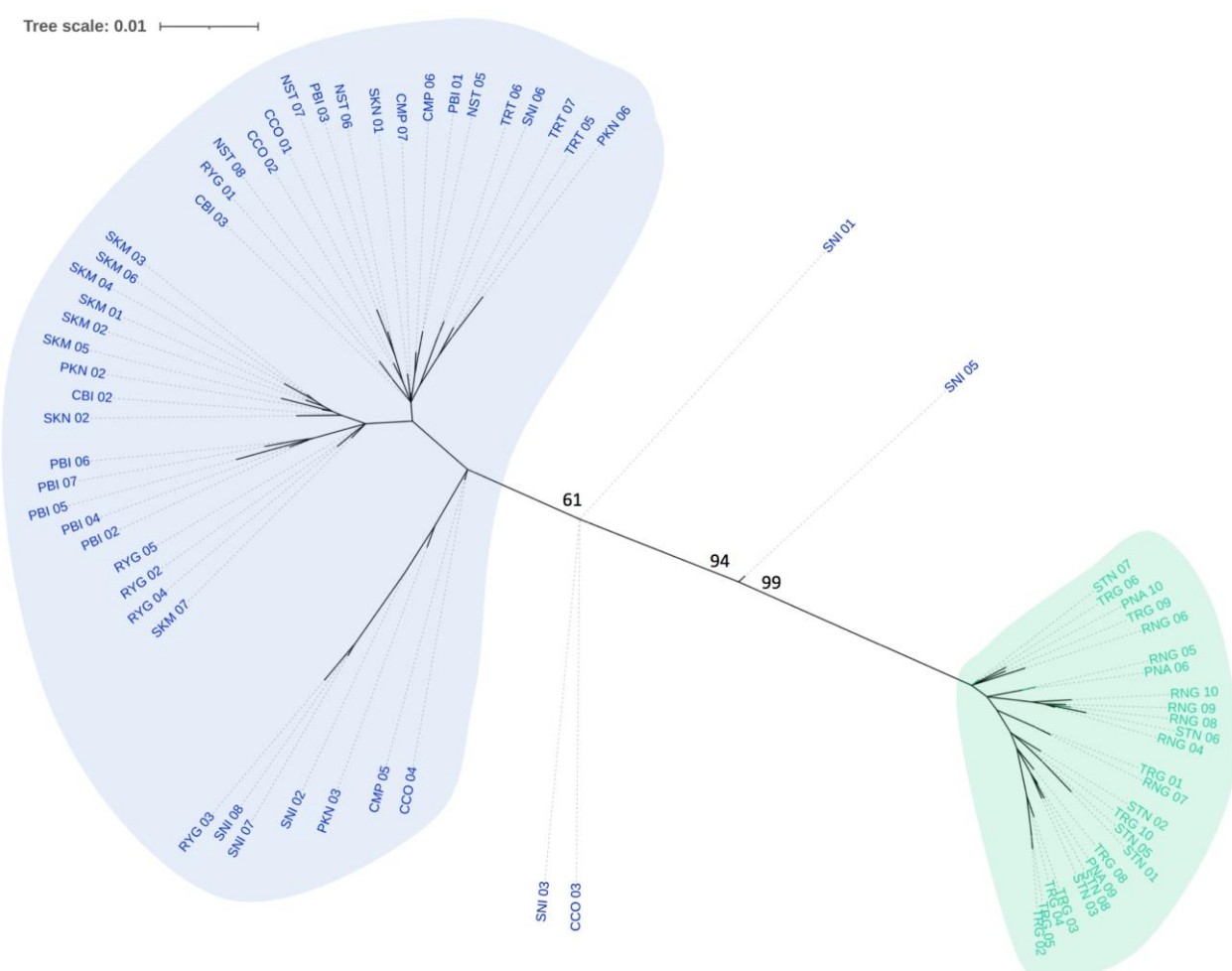

**Figure 3.** Unrooted phylogenetic tree inferred by maximum likelihood analysis of 73 *B. gymnorhiza* accessions using 1519 SNP markers. Blue and green texts in the ML tree indicate two different sampling sites in the Gulf of Thailand and the Andaman Sea sides, respectively.

*3.3. Population Genetic Structure and PCA*

The genetic structure of the 73 *B. gymnorhiza* accessions based on the 1519 SNP markers was evaluated using Bayesian clustering and PCA (Figure 4). Based on Bayesian clustering in STRUCTURE, delta *K* showed a maximum peak at *K* = 2 (the optimal number of clusters), with a small peak again at *K* = 4 (Figure 4A). The distribution probabilities of clustering the accessions when *K* = 2 and *K* = 4 were plotted (Figure 4B). When *K* = 2, the accessions were assigned to the two genetic clusters (blue and green), corresponding to the east and west coastlines of Thailand. The accessions from Chachoengsao (CCO) and Surat Thani (SNI) were admixed with the admixture proportions in both the Gulf of Thailand and Andaman populations. SNI_05 collected from the Gulf of Thailand had the highest proportion of admixture from the Andaman population (52%). When *K* = 4, several accessions from Chonburi (CBI), Chumphon (CMP), Rayong (RYG), Phetchaburi (PBI), Surat Thani (SNI), and Trang (TRG) were also admixed. Based on PCA, the first two principal components of the PCA strongly support the two main genetic clusters (Figure 4C). The percentage of variance explained accounted for 34.9% and 12.8% of the total variation in PC1 and PC2, respectively. Additionally, the accessions from RYG, PBI, SNI, and TRG were far from

their clusters, indicating a high genetic variation. In the Gulf of Thailand cluster, all PBI accessions and some RYG accessions formed a subcluster as unique individuals when *K* = 4 in STRUCTURE (Figure 4B,C). In the Andaman cluster, the subcluster in the upper, right quadrant contained only TRG accessions in the Andaman population as unique individuals when *K* = 4 in STRUCTURE (Figure 4B,C). SNI_05 was in the middle of the PCA plot, indicating an admixed accession.

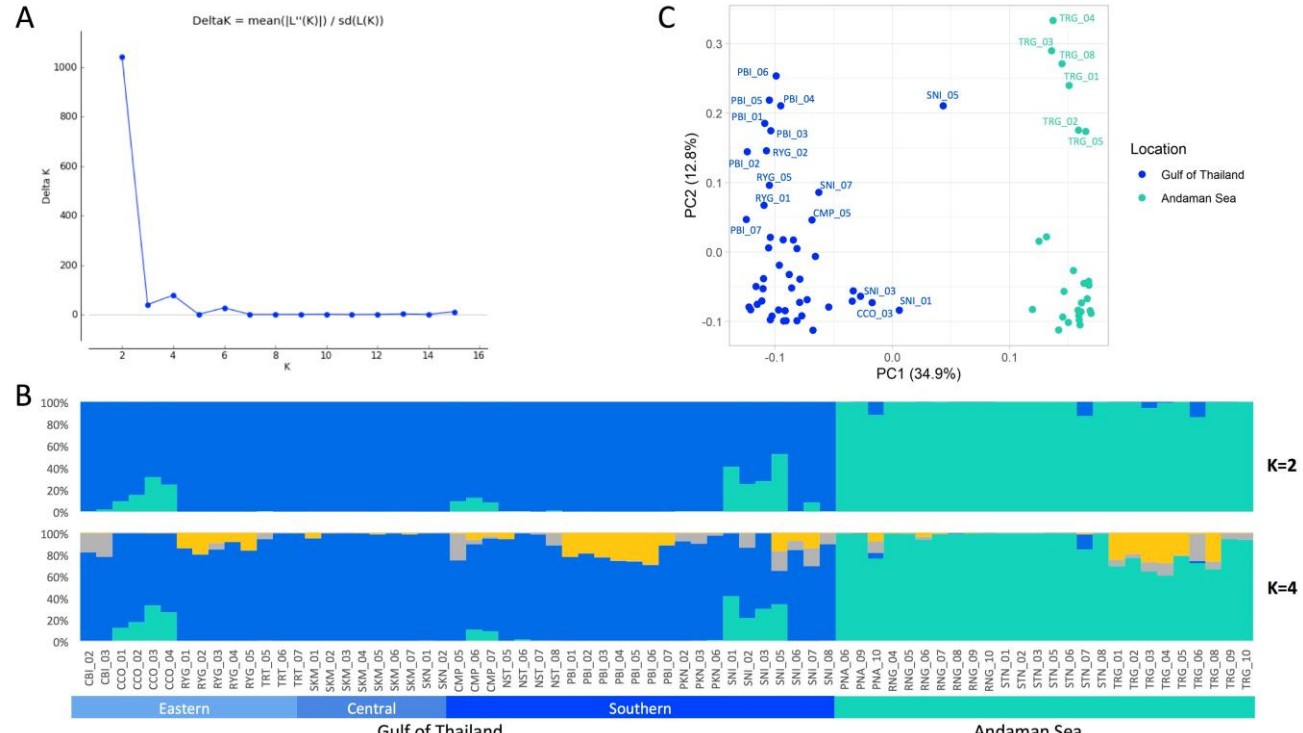

**Figure 4.** The population structure of 73 *B. gymnorhiza* accessions. (**A**) Delta *K* plot representing the best cluster number assumed *K* from 1 to 16 in the STRUCTURE analysis. (**B**) Population structure analysis, *K* = 2 and *K* = 4. Each accession is represented by a single column. Colors represent different clusters. (**C**) The principal component analysis plot of the first two PCs.

*3.4. Genetic Diversity and Differentiation*

The parameters show a moderate level of genetic diversity in *B. gymnorhiza* (Table 1; Table S4). The average *N*a, *N*e, *I*, *H*o, and *H*e values were 2.463 (2.417–2.510), 1.537 (1.517–1.557), 0.522 (0.515–0.529), 0.397 (0.383–0.410), and 0.317 (0.311–0.323), respectively. The average percentage of polymorphic loci (PPL) was 99.51%. The PPL of the accessions from the Gulf of Thailand coast was 100.00%, whereas that from the Andaman Sea coast was 99.01%. The inbreeding coefficient for all accessions and the accessions in each cluster was a negative value, indicating more heterozygous genotypes.

**Table 1.** Genetic diversity parameters of *B. gymnorhiza* (L.) Savigny in each of the two subpopulations and all accessions based on 1519 SNPs.

| Population | N | *Na* | *Ne* | *I* | *Ho* | *He* | PPL (%) | *F* |
|---|---|---|---|---|---|---|---|---|
| Gulf of Thailand | 46 | 2.510 ± 0.013 | 1.517 ± 0.008 | 0.515 ± 0.005 | 0.383 ± 0.007 | 0.311 ± 0.004 | 100.00 | −0.135 ± 0.009 |
| Andaman Sea | 27 | 2.417 ± 0.013 | 1.557 ± 0.009 | 0.529 ± 0.005 | 0.410 ± 0.008 | 0.323 ± 0.004 | 99.01 | −0.157 ± 0.010 |
| Overall | 73 | 2.463 ± 0.009 | 1.537 ± 0.006 | 0.522 ± 0.004 | 0.397 ± 0.005 | 0.317 ± 0.003 | 99.51 | −0.146 ± 0.007 |

Based on the two main genetic clusters from STRUCTURE, the genetic differentiation of the populations based on AMOVA showed that 16.05% and 83.95% of the total variation occurred among populations and within populations, respectively (Table 2). The

significant $F_{ST}$ value between the subpopulations was 0.16 ($p < 0.001$), revealing a high genetic differentiation.

**Table 2.** Analysis of molecular variance (AMOVA) using 1519 SNPs of the genetic variation among and within two subpopulations of 73 *B. gymnorhiza* accessions.

| Source of Variation | df | Sum of Squares | Variance Components | Percentage of Variation | F-Statistics |
|---|---|---|---|---|---|
| Among populations | 1 | 2772.27 | 37.83 | 16.05 | $F_{ST}$ = 0.16 *** |
| Within populations | 144 | 28,495.58 | 197.89 | 83.95 | |
| Total | 145 | 31,267.85 | 235.71 | | |

Note: df = degree of freedom; *** = statistical significance at $p < 0.001$.

## 4. Discussion

### 4.1. Genetic Relationship and Genetic Structure

To understand the relationships among the *B. gymnorhiza* accessions in Thailand, an unrooted ML tree based on SNP markers revealed two genetic clusters (the cluster of the Gulf of Thailand and the cluster of the Andaman Sea), concordant with the result of PCA and a structure analysis. It is also consistent with previous studies on *Bruguiera* species [32,34,36,37,39] and other mangrove species [33,62–67]. For example, based on nuclear and chloroplast regions, *B. gymnorhiza* individuals collected from India, Indonesia, Malaysia, Micronesia, Thailand, and the southern islands of Japan formed three clusters: (1) the eastern region of the Malay Peninsula, including Indonesia, Malaysia, Micronesia, Thailand, and the southern islands of Japan; (2) the eastern coast of India and the western coast of the Malay Peninsula; and (3) the western coast of India [36] in which the two clusters agreed with our result. Using nuclear gene regions, ten *B. gymnorhiza* populations in the IWP region were divided into two genetic clusters: the east and west clusters [37]. Using ISSR, the *Ceriops tagal* population on the eastern coast of Thailand resembled the population from South China more than the population on the west coast of Thailand [62]. Two subpopulations of *C. tagal* and *Rhizophora apiculata* in Thailand corresponding to the eastern and western coasts of Thailand were reported using SNP markers [33,67]. These support the land barrier hypothesis of the Malay Peninsula promoting genetic divergence between the eastern and western sides [68]. In addition, genetic admixture was found in *B. gymnorhiza* and other *Bruguiera* species, such as *B. cylindrica* and *B. parviflora* [32,34], as well as in other mangrove species, such as *R. apiculata* [33]. Genetic admixture could increase the opportunity for mangrove species to adapt to local environments.

### 4.2. Genetic Diversity and Genetic Differentiation

The genetic diversity of the *B. gymnorhiza* accessions (mean $H$o = 0.397 and mean $H$e = 0.317) was relatively moderate compared with the published mangrove data of *B. cylindrica* ($H$o = 0.289 and $H$e = 0.257: a low genetic diversity) and *R. apiculata* ($H$o = 0.478 and $H$e = 0.360: a moderate genetic diversity) based on SNP markers [33,34]. To estimate the levels of genetic diversity, SNPs have more accuracy than microsatellites, which mostly assessed a higher diversity [29,30,69]. It seems that the genetic diversity of the *B. gymnorhiza* population based on SNP markers was higher than that of several *Bruguiera* populations in the IWP region based on various genetic markers [14,35–37,39,40]. For example, a very low variation in *B. gymnorhiza* ($H$o = 0.025 and $H$e = 0.035) was found in six populations from the southwestern islands of Japan based on allozyme data [35]. Using nuclear and chloroplast microsatellite markers, the low genetic diversity of *B. gymnorhiza* ($H$o = 0.314 and $H$e = 0.408) was observed in nine populations from Iriomote Island of the Ryukyu Archipelago in Japan [40]. The low genetic diversity of *B. gymnorhiza* ($H$o = 0.316 and $H$e = 0.356) was also reported in nine populations along the coastlines of South China based on chloroplast microsatellite markers [39]. In addition, the genetic diversity of *B. gymnorhiza* in this study was lower than that of other terrestrial plants [70–72]. For example,



*Trigonobalanus doichangensis*, an endangered plant in China, had a high genetic diversity (*H*o = 0.557 and *H*e = 0.306) based on SNP markers [72]. Based on microsatellite markers, *Eucalyptus urophylla*, an economically important tropical forest tree, had a moderate-to-high genetic diversity (*H*e = 0.510–0.720), and *Shorea obtusa*, a deciduous tropical tree, had a high genetic diversity (*H*e = 0.664) [70,71]. Generally, mangrove plants have a low genetic diversity [26,27,29,30,32,33,62–66,73] that is mainly caused by population size reduction, habitat fragmentation, population bottlenecks, limited propagule dispersal, and climate fluctuations [64,73–75]. In Thailand, approximately 87% of the original mangrove areas have been lost compared to the mangrove forest in the 1960s [76], leading to less genetically diverse mangrove species. A low genetic diversity may have an impact on the long-term survival of mangrove populations.

A high level of genetic differentiation between two *Bruguiera gymnorhiza* subpopulations ($F_{ST}$ = 0.16, $p < 0.001$) was observed. This level of genetic differentiation was similar to that of other *B. gymnorhiza* populations ($F_{ST}$ = 0.16–0.89, $p < 0.001$) in the IWP region [37,39]. Indeed, pollination and seed dispersal in *B. gymnorhiza* appear to be limited [14–16,36,77]. *B. gymnorhiza* is pollinated by birds [14–16,77]. Seed dispersal in *B. gymnorhiza* is limited within an ocean [36]. Thus, significant genetic divergence was observed in the *B. gymnorhiza* accessions on different oceans.

The inbreeding coefficients of *B. gymnorhiza* (*F* = −0.162 to −0.146) were negative, indicating an excess of heterozygotes. They were similar to those of other mangroves, such as *B. cylindrica* (*F* = −0.229 to −0.162) and *R. apiculata* (*F* = −0.258 to −0.140), based on SNP markers [33,34]. In contrast, the inbreeding coefficients of *B. gymnorhiza* in South China based on allozyme markers (*F* = −0.076 to 0.132) and nuclear and chloroplast microsatellite markers (*F* = −0.096 to 0.401) varied from negative to positive values [14,39]. Positive inbreeding coefficients were observed in the southwest Islands of Japan based on allozyme markers (*F* = 0.051 to 0.462), in Iriomote island of Japan based on nuclear microsatellites (*F* = 0.016 to 0.422), and in the IWP region based on nuclear genes (*F* = 0.292) [35,37,40]. These results show that SNPs presented more heterozygotes than other molecular markers. In addition, the AMOVA results showed that the most genetic variation occurred within the *B. gymnorhiza* populations, concordant with other studies in *Bruguiera* species [25,34]. Indeed, most genetic variation occurs within populations in outcrossing plants, whereas self- and mixed-mating plants preserve the majority of genetic variation among populations [78,79]. The *F* values based on SNPs suggest that the *B. gymnorhiza* populations in Thailand may either be predominantly outcrossing, or that homozygous individuals produced by inbreeding may be strongly selected against, concordant with previous reports [9,14,15]. A higher genetic variation within populations would be beneficial, allowing them to evolve and adapt to cope with local environments, particularly climate changes.

## 5. Conclusions

The population structure of *Bruguiera gymnorhiza* populations along the coastlines of Thailand was shown to be strongly structured. Two main genetic clusters, the Gulf of Thailand and the Andaman Sea coasts, were identified based on a phylogenetic tree analysis, a STRUCTURE analysis, and PCA, suggesting that the genetic dispersal of *B. gymnorhiza* in Thailand is strongly driven by the geographic barrier of the Malay Peninsula. Although two distinct clusters were evident, some substructuring within the two coastal populations was also evident—for example, the TRG accessions were distinct from other Andaman Sea accessions. Some of these accessions were in admixture. *B. gymnorhiza* had a moderate genetic diversity and a high genetic differentiation. Based on AMOVA, 83.95% of the variation was found within populations, and 16.05% of the variation was found among populations. The results provide a clear understanding of the genetic structure and diversity of *B. gymnorhiza* in Thailand for mangrove conservation and management that support the preservation of genetic diversity and avoid mixing genetically distinct populations.

**Supplementary Materials:** The following supporting information can be downloaded at https://www.mdpi.com/article/10.3390/f14040693/s1, Table S1: List of collection sites and sample sizes of *B. gymnorhiza*; Table S2: Statistics for the mapping of *B. gymnorhiza*; Table S3: List of 1519 SNPs and their characteristics in *B. gymnorhiza*; Table S4: Summary of genetic diversity parameters for two main genetic clusters of 73 *B. gymnorhiza* accessions based on 1519 SNPs.

**Author Contributions:** Conceptualization, P.R.-a., W.P. and S.T.; methodology, P.R.-a., W.P. and S.T.; formal analysis, P.R.-a. and C.S.; investigation, D.S. and P.W. (Pitchaporn Waiyamitra); resources, C.M., P.W. (Poonsri Wanthongchai) and P.C.; writing—original draft preparation, P.R.-a.; writing—review and editing, P.R.-a., W.P. and S.T.; visualization, P.R.-a.; supervision, W.P. and S.T.; funding acquisition, S.T. All authors have read and agreed to the published version of the manuscript.

**Funding:** This research was funded by the National Science and Technology Development Agency, Thailand, No. P1952261.

**Institutional Review Board Statement:** Not applicable.

**Informed Consent Statement:** Not applicable.

**Data Availability Statement:** SNP data are available in Table S3.

**Acknowledgments:** The authors would like to thank the researcher team from Thailand Mangrove Forest Research Center for sample collection.

**Conflicts of Interest:** The authors declare no conflict of interest.

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
