# Peer review of "Elucidating SNP-Based Population Structure and Genetic Diversity of Bruguiera gymnorhiza (L.) Savigny in Thailand"

_forests, doi:10.3390/f14040693_

Round 1

Author Response

Dear Review,

We would like to express our gratitude and appreciation to the reviewers for providing insightful comments concerning this manuscript. We have revised our manuscript and addressed all comments received. Our point-by-point responses are presented in the attached file.

Best regards,

Panthita

Reviewer 2 Report

The authors present a straightforward study on the population structure of the Thai subpopulation of mangrove species Bruguiera gymnorhiza. The study is technically well executed and the paper is clear. However, it needs some revision before being acceptable in Forests.  My recommendations include:

1.     Include information about the breeding system in the Introduction. This is finally included at line 292 but it should be up front. The study finds that there is a strongly negative FIS in both populations. What does this say about the breeding system of the population? Please compare this result to other studies. I think this is an important piece of commentary because it might suggest that either the species does not have a mixed mating system or that there is very strong selection against homozygotes.

2.     The authors claim that the species has low genetic diversity. Please provide some definitions of low, medium and high genetic diversity and compare the results with other forest species that have medium or high diversity. Also, discuss the impact of marker type on diversity measures. These things could be partly in the Introduction, partly in Discussion.

3.     The authors do not provide any justification for selecting K= 10 Structure groups. Usually, the number of groups is the number of putative subpopulations plus one. It is possible that more K groups should have been used – though I am not suggesting that this would change the optimal K which appears to be either 2 or 3.

4.       The PCA clearly shows three clusters. What is the cluster in the lower, right quadrant (Andaman Sea)? Is there any geographic basis for this cluster? Does it correspond to the individuals that are admixed in the K=3 graph? It looks like it might correspond to the lower branch in the Andaman cluster of the phylo tree. Please discuss this. It shows that while there are two MAIN groups, there is some substructure underneath this. Do not be over-reliant on the Delta K method to identify the “true” number of subpopulations. It is often not completely effective and only detects the largest subpopulation divisions. I suggest you could look at https://doi.org/10.1007/s00468-014-1092-0  Chong et al. on Chukrasia to see an approach to examining Structure results but I do not suggest this paper should be cited.

5.     I am not sure what the C E and S coding in the PCA graph mean. Please explain this in the caption and in the Results

6.     The paper is generally well written but I have attached a marked-up version of the PDF with some corrections. There are also some additional technical comments marked up.

Although I have recommended "major" revision I think the authors should be able to accomplish the changes quite easily and I look forward to seeing the revised version.

Author Response

(The authors gave the same response as above.)

Reviewer 3 Report

In this study, the authors analyzed the population structure and genetic diversity of Bruguiera gymnorhiza in Thailand by SNP marker. I salute the authors for their intention. Regionally species like B. gymnorhiza need our attention. The subject is interesting. This study is useful for future Mangroves studies. It provides good data on population structure and genetic diversity of B. gymnorhiza in Thailand for future studies. The concept of this study is generally correct. The author's introduction is well written and the experimental design is good. Here, I'd like to offer some personal advice that I hope will be useful to authors.

- The keywords words should not be repeated with the words included in the title.

- Authors should check the manuscript throughout, which contains minor errors, such as "Indonesia, Malaysia, Micronesia, Thailand and the". "and" needs to be preceded by a ",".

- Materials and Methods. Some programs include a version; others don't. The software used should ideally provide a citation or website.

- The manuscript mentions Bruguiera gymnorhiza as likely to be confused with other Bruguiera species. A previous article on the conservation of Phoebe species provided the idea that the complete chloroplast genome is used to discriminate between species. The manuscript may even be enriched by using the reference:

Shi W, Song W, Chen Z, et al. Comparative chloroplast genome analyses of diverse Phoebe (Lauraceae) species endemic to China provide insight into their phylogeographical origin[J]. PeerJ, 2023, 11: e14573.

Author Response

(The authors gave the same response as above.)

Reviewer 4 Report

This is my review of the paper entitled "Elucidating SNP-based population structure and genetic diversity of Bruguiera gymnorhiza in Thailand". The authors of this study investigated the populations diversity and structure of Bruguiera gymnorhiza that thrive along Thailand's coastlines by utilizing RAD-seq to obtain high-quality SNPs. The study revealed low genetic diversity among the accessions, but high genetic differentiation among different groups. These findings could potentially provide valuable insights for the management of mangrove ecosystems.

Major comments:

I would like to raise a concern regarding the lack of consideration of Linkage Disequilibrium (LD) blocks in the study, which can potentially introduce bias in the inference of population structure and phylogenetic tree. To address this issue, I suggest that the authors pre-process their data by pruning the SNPs using Plink (--indep-pairwise) or VCFTools (--thin) to remove any potential effect of LD. If this results in a small number of SNPs, the authors could loosen the missing rate of the SNP filtering, as allowing only 5% missing may be too strict for RAD-seq data.

Given that the authors utilized RAD-seq to obtain high-quality SNPs, I suggest that they compare RAD-seq with other sequencing methods and highlight its advantages. E.g. RAD-seq has been shown to be a cost-effective and efficient method for SNP discovery and genotyping, especially in non-model organisms where whole-genome sequencing may be impractical.

The authors calculated genetic diversity using the Minor Allele Frequency (MAF) filtered SNPs and compared the results to those obtained from other methods, such as allozyme and microsatellite analysis. However, it may be worth noting that using MAF-filtered SNPs may not provide a comprehensive estimate of genetic diversity, as it only considers a subset of the total genetic variation. I would like to suggest that the authors consider using a more suitable parameter, such as π, to represent genetic diversity. Calculating π based on all SNPs, including non-polymorphic sites, provides a more comprehensive and accurate estimate of genetic diversity. Alternatively, I recommend that the authors provide examples or previous studies to support their claim that the genetic diversity of Bruguiera gymnorhiza is low. This will strengthen the argument and provide more credibility to the conclusions drawn from the study

Minor comments:

Lines 125-126: I don't think GATK has the capability to perform demultiplexing, so I suggest double-checking this information.

Lines 130-132: Consider filtering SNPs based on parameters such as read depth (DP) and genotype quality (GQ). This can help to remove potential sequencing errors and low-quality variants. You can neglect it if it is not applicable for this study.

Regarding Figure 2, I noticed that the bucketed ranges in the histogram may be incorrect since some ranges, such as 0.05-0.06 and 0.10-0.11, appear to be missing. Additionally, since minor alleles were filtered out, the range of 0.00-0.05 may not be necessary. Could you please clarify which type of diversity is being shown in Figure 2A?"

Lines 202-204: I'm having trouble understanding this sentence. Could you please rephrase or clarify what you mean so that I can better understand it?

Author Response

(The authors gave the same response as above.)

Round 2

Reviewer 2 Report

I am satisfied that the authors have made a significant attempt to address nearly all of the points I raised in review of the original MS in addition to points raised by other reviewers. I have attached a marked-up version of the PDF with some additional suggestions for improved grammar and a couple of minor points. I recommend acceptance once the journal editor has considered that these minor changes have been made without need for further peer review with respect to the points I have raised.

Author Response

We would like to thank you for all your valuable comments.

We have edited the manuscript.  Our point-by-point responses are presented in the attached comment file.

Best regards,

Panthita

Reviewer 4 Report

The authors have provided clear responses to the points raised in my review. However, it appears that the main point of my review may have been missed. While unpruned data can potentially introduce bias in the inference of population structure and phylogenetic tree, it should not affect diversity estimation. One solution could be to use different datasets for different analyses, such as using the pruned dataset for population structure and phylogenetic analyses, while using the whole dataset for diversity estimation. If the other reviewers are satisfied with the current version, the authors may choose to keep it as is.

Regarding Figure 2, I noticed that the dataset was filtered using a MAF filter, which removes values with MAF <=0.05. It appears that the histogram between the values of 0.05 to 0.06 is missing. 

It seems that the data availability section may be missing from the manuscript.

Author Response

(The authors gave the same response as above.)
